# Leveraging Causal Graphs for Blocking in Randomized Experiments

## Abstract

*Randomized experiments* are often performed to study the causal effects of interest. *Blocking* is a technique to precisely estimate the causal effects when the experimental material is not homogeneous. It involves stratifying the available experimental material based on the covariates causing non-homogeneity and then randomizing the treatment within those strata (known as *blocks*). This eliminates the unwanted effect of the covariates on the causal effects of interest. We investigate the problem of finding a *stable* set of covariates to be used to form blocks, that minimizes the variance of the causal effect estimates. Using the underlying causal graph, we provide an efficient algorithm to obtain such a set for a general *semi-Markovian* causal model.

## 1 Introduction

### 1.1 Motivation

Studying the *causal* effect of some variable(s) on the other variable(s) is of common interest in social sciences, computer science, and statistics. However, a mistake that people usually make is, confusing the causal effect with an *associational* effect. For instance, if high levels of bad cholesterol and the presence of heart disease are observed at the same time, it does not mean that the heart disease is caused by the high levels of bad cholesterol. The question is then how do we get to know if at all a variable causes the other? If the answer is yes, then what is the direction (positive or negative) and what is the magnitude, of the causal effect? Fisher (1992) provided the framework of *randomized experiments* to study the causal effect, where the variable whose causal effect is to be studied (known as *treatment* or cause), is randomized over the available experimental material (like humans, rats, agricultural plots, etc.) and changes in the variable on which the causal effect is to be studied (known as *response* or effect), are recorded. A statistical comparison of values of the response with or without the treatment can therefore be done to study the existence, direction, and magnitude of the cause-effect relationship of interest.

Randomized experiments work on three basic principles, viz., *randomization*, *replication*, and *local control*. Randomization states that the assignment of the treatment has to be random, replication states the treatment should be given to multiple but homogeneous units, i.e., there are multiple observations of the effect variable for both with and without the treatment. Therefore, as long as the entire experimental material is homogeneous (for instance, the fertility of all the agricultural plots is the same, the responsiveness of all the humans is the same for the drug, etc.), then a 'good' randomized experiment can be carried out using the first two principles, viz., randomization and replication, which gives rise to *completely randomized design* (CRD). However, when the entire experimental material is not homogeneous, i.e., some attributes of experimental units (known as *covariates*) differ from each other, then the causal effect may get influenced by the covariates causing non-homogeneity (like fertility, responsiveness, etc). The remedy to this problem is the third principle of randomized experiments, i.e., local control (also known as *blocking*), which states to stratify the entire experimental material based on that covariate(s) causing non-homogeneity, and then randomize the treatment within those strata to eliminate the effect of the covariates. These strata are called *blocks*, and this third principle along with the first two give rise to *randomized block design* (RBD). Blocking tries to control/eliminate the variability in the response attributed through the covariates and leads to a more *precise*

estimation of the causal effect of interest. Precision is defined as the inverse of the variance of the estimate of the causal effect of interest.

In this paper, we focus on the problem of deciding which covariates to be used for forming the blocks while performing a randomized experiment. We consider a non-parametric setting and assume that we have access to the underlying causal structure (Pearl, 2009). We provide an efficient algorithm to obtain a stable set of covariates to be used to form blocks for a general semi-Markovian causal model.

### 1.2 Literature Review

Statistics literature (Yates, 1935), (Kempthorne, 1952), (Kempthorne, 1955), (Cochran & Cox, 1948) presents profound discussions on relative efficiency (ratio of the variances of causal effects) of RBD over CRD. It can be seen that in general, RBD is always more or at least as efficient as CRD. Therefore, the natural question is how do we do blocking in an intelligent manner such that maximum gain in precision can be attained? More specifically, we are interested in answering the following question. Can we decide which covariates to be used for forming blocks using the causal structure (diagram) of all the variables, viz., treatment, response, and covariates (observed and unobserved) provided the causal structure is given?

Cinelli et al. (2020) provides a good discussion on good and bad controls for different causal graphs. However, they do not provide a general algorithm to address the problem for a general causal graph. In contrast, we provide an efficient algorithm to obtain the set of covariates to be used for forming blocks for a general semi-Markovian causal model.

In this paper, we focus on deciding what covariates should form the basis of forming blocks. However, a problem that follows our problem is, how to efficiently form blocks by grouping the experimental units which are close to each other based on the chosen covariates. Moore (2012); Higgins et al. (2016) focus on the latter problem.

### 1.3 Contribution

In this paper, we formalize the problem of finding a *stable* set of covariates to be used to form blocks, that minimizes the variance of the causal effect estimates. We leverage the underlying causal graph and provide an efficient algorithm to obtain such a set for a general *semi-Markovian* causal model.

### 1.4 Organization

The rest of the paper is organized as follows. Section 2 provides some background and formulates the problem of interest. Section 3 discusses our methodology that includes motivating examples, key observations, important lemmas, main results, and final algorithm. Section 4 concludes the paper and provides future directions.

## 2 Background and Problem Formulation

In this section, we provide some background and formulate the problem of interest in this paper.

Let $\mathcal{V}$ be the set of observed/endogenous random variables and $\mathcal{U}$ be the set of unobserved/exogenous random variables. Let $\mathcal{G}$ be a directed acyclic graph that depicts the causal relationships among the random variables in $\mathcal{U}$ and $\mathcal{V}$. We are interested in studying the causal effect of $X \in \mathcal{V}$ (treatment) on $Y \in \mathcal{V}$ (response). Studying the causal effect of $X$ on $Y$ means learning the (interventional) probability distribution, $\mathbb{P}(Y|do(X))$ (Pearl, 2009).

### 2.1 Gain in Precision due to Blocking

Consider a simple setting where $X$ is an indicator random variable defined as

$$X = \begin{cases} 1, & \text{if the treatment is applied,} \\ 0, & \text{otherwise,} \end{cases} \tag{1}$$

and $Y$ is real-valued random variable.

Let there be $n$ experimental units available to study the causal effect of $X$ on $Y$ by performing a randomized experiment. Let $\mathcal{Z}$ denote the set of covariates that vary across the experimental units.

We want to demonstrate the gain in precision due to blocking. Thus, we first ignore these covariates and perform a CRD as follows. We randomize $X$ over the available experimental units, i.e., randomly assign a unit to either have the treatment ($X = 1$) or not ($X = 0$) and then observe the response. Let $(X_i, Y_i)$ denote the treatment and response pair for the $i$th experimental unit. Define $I_x := \{i : X_i = x\}$ and $n_x := |I_x|$.

For demonstrating the gain in precision due to blocking, we focus on estimating $\mathbb{E}(Y|do(X))$. For simplicity, define $Y(x) := Y|do(X = x)$. For our simple setting, we have only two possibilities of the that conditional expectation, viz. $\mathbb{E}(Y(1))$ and $\mathbb{E}(Y(0))$. Similar to Heckman (1992), Clements et al. (1994), and Heckman & Smith (1995), we define the *effect of treatment* (ET) in the population as

$$\beta := \mathbb{E}(Y(1)) - \mathbb{E}(Y(0)). \tag{2}$$

A natural non-parametric estimator of $\mathbb{E}(Y(x))$ is the corresponding sample average, given as

$$\bar{Y}(x) := \frac{1}{n_x} \sum_{i \in I_x} Y_i. \tag{3}$$

Therefore, we can estimate $\beta$ as

$$\hat{\beta} = \bar{Y}(1) - \bar{Y}(0) \tag{4}$$

It can be shown that $\mathbb{E}(\hat{\beta}) = \beta$, i.e., $\hat{\beta}$ is *unbiased* for $\beta$. The variance of $\hat{\beta}$ is given as follows.

$$\text{Var}(\hat{\beta}) = \mathbb{E}_{\mathcal{Z}} \left( \frac{\text{Var}(Y(1)|\mathcal{Z})}{n_{1,z}} + \frac{\text{Var}(Y(0)|\mathcal{Z})}{n_{0,z}} \right) + \mathbb{E}_{\mathcal{Z}}(\beta(\mathcal{Z}) - \beta)^2, \tag{5}$$

where $\beta(\mathcal{Z}) := \mathbb{E}(Y(1)|\mathcal{Z}) - \mathbb{E}(Y(0)|\mathcal{Z})$ is the $\mathcal{Z}$-specific causal effect. For proofs of unbaisedness and variance of $\hat{\beta}$, see Appendix A.1.

We now perform an RBD as follows. We first stratify the experimental units such that units with each stratum (known as a block) are identical, i.e., the covariates in $\mathcal{Z}$ remain the same with a block. We next randomize $X$ over the experimental units within each block and then observe the response. Let $(X_i, Y_i, \mathcal{Z}_i)$ denote the treatment, response, and covariates triplet for the $i$th experimental unit. Define $I_{x,z} := \{i : X_i = x, \mathcal{Z}_i = z\}$ and $n_{x,z} := |I_{x,z}|$.

Note that ET defined in equation 2 can be re-written as (Petersen et al., 2006)

$$\beta = \mathbb{E}_{\mathcal{Z}}(\mathbb{E}(Y(1)|\mathcal{Z}) - \mathbb{E}_{\mathcal{Z}}(\mathbb{E}(Y(0)|\mathcal{Z}), \tag{6}$$

$$= \mathbb{E}_{\mathcal{Z}}(\mathbb{E}(Y(1)|\mathcal{Z}) - \mathbb{E}(Y(0)|\mathcal{Z})), \tag{7}$$

$$= \sum_{\mathcal{Z}} (\mathbb{E}(Y(1)|\mathcal{Z}) - \mathbb{E}(Y(0)|\mathcal{Z})) \mathbb{P}(\mathcal{Z}) \tag{8}$$

A natural non-parametric estimator of $\mathbb{E}(Y(x)|\mathcal{Z})$ is the corresponding sample average, given as

$$\bar{Y}(x)|\mathcal{Z} := \frac{1}{n_{x,z}} \sum_{i \in I_{x,z}} Y_i \tag{9}$$

Define $n_z := n_{1,z} + n_{0,z}$. $\mathbb{P}(\mathcal{Z})$ can be estimated as

$$\hat{\mathbb{P}}(\mathcal{Z}) := \frac{n_z}{n} \tag{10}$$

Therefore, we can estimate $\beta$ as

$$\hat{\beta}_{\mathcal{Z}} = \frac{1}{n} \sum_{\mathcal{Z}} n_z \left( \bar{Y}(1)|\mathcal{Z} - \bar{Y}(0)|\mathcal{Z} \right) \tag{11}$$

It can be shown that $\mathbb{E}(\hat{\beta}_{\mathcal{Z}}) = \beta$, i.e., $\hat{\beta}_{\mathcal{Z}}$ is *unbiased* for $\beta$. The variance of $\hat{\beta}_{\mathcal{Z}}$ is given as follows.

$$\text{Var}(\hat{\beta}_{\mathcal{Z}}) = \mathbb{E}_{\mathcal{Z}} \left( \frac{\text{Var}(Y(1)|\mathcal{Z})}{n_{1,z}} + \frac{\text{Var}(Y(0)|\mathcal{Z})}{n_{0,z}} \right) + \mathbb{E}_{\mathcal{Z}} \left( \sum_{\mathcal{Z}} \beta(\mathcal{Z}) \hat{\mathbb{P}}(\mathcal{Z}) - \beta \right)^2, \tag{12}$$

For proofs of unbiasedness and variance of $\hat{\beta}_{\mathcal{Z}}$, see Appendix A.2.

If $\hat{\mathbb{P}}(\mathcal{Z})$ is as good as $\mathbb{P}(\mathcal{Z})$ then $\sum_{\mathcal{Z}} \beta(\mathcal{Z}) \hat{\mathbb{P}}(\mathcal{Z}) = \mathbb{E}_{\mathcal{Z}}(\beta(\mathcal{Z})) = \beta$ then the second term on the right in equation 12 is zero, and by comparing equation 5 and equation 12, we observe that $\text{Var}(\hat{\beta}_{\mathcal{Z}}) \leq \text{Var}(\hat{\beta})$, i.e., blocking improves the precision of the estimate of the population average causal effect, $\beta$.

## 2.2 Challenges with Blocking

We observed that in experimental studies under non-homogeneity of the experimental units, blocking improves the precision of causal effect estimation. However, the practical difficulty with blocking is that depending on the number of covariates and the number of distinct values (denoted as $v(\cdot)$) of each covariate, the number of blocks can be very large. For covariates in the set $\mathcal{Z}$, we need to form $\prod_{Z \in \mathcal{Z}} v(Z)$ different blocks. For example, if we want to study the effect of a drug on curing some heart disease where the subjects under consideration have the following attributes (which can potentially affect the effect of the drug).

1. Gender: Male and Female, i.e., $v(\text{Gender}) = 2$,

2. Age: <25, 25-45, 45-65, >65, i.e., $v(\text{Age}) = 4$,

3. Weight: Underweight, Normal, Overweight, Obese I, Obese II, i.e., $v(\text{Weight}) = 5$,

4. Blood Pressure: Normal, Prehypertension, Hypertension I, Hypertension II, Hypertensive Crisis, i.e., $v(\text{Blood Pressure}) = 5$, and

5. Bad Cholesterol: Optimal, Above Optimal, Borderline High, High, Very High, i.e., $v(\text{Bad Cholesterol}) = 5$.

Thus, we need to form $2 \times 4 \times 5 \times 5 \times 5 = 1000$ blocks. Performing a randomized experiment with a large number of blocks can be very costly. Sometimes, it may not be feasible as the number of blocks can be larger than the number of subjects. This would cause some of the blocks to be empty. For instance, there may not be any male subjects under the age of 25 who are Obese II with a Hypertensive Crisis and Optimal Cholesterol level.

Other than the economic aspects and some blocks being empty, there are reasons why some variables should never be used for forming blocks. See Section 3.4 for details.

## 2.3 Problem Statement

So far we observed that blocking improves the precision of causal effect estimation. However, in situations when the number of blocks to be formed is very large, blocking becomes costly and/or infeasible. One possible way to reduce the number of blocks is to form blocks using some but not all covariates. But the question is which covariates should be preferred over others while forming blocks? One possible way is to select the

(smallest) set of covariates that leads to a maximum precision in causal effect estimation. In the context of estimating the effect of treatment in the population, $\beta$, (discussed in Section 2.1), it means finding a smallest set, $\mathcal{Z}$, that minimizes $\mathrm{Var}\,(\hat{\beta}_{\mathcal{Z}})$.

We next formalize the problem of interest in this paper as follows. We are given a directed acyclic causal graph, $\mathcal{G}$, that depicts the causal relationships among some observed variables, $\mathcal{V}$, and unobserved variables, $\mathcal{U}$. We are interested in studying the causal effect of $X \in \mathcal{V}$ (treatment) on $Y \in \mathcal{V}$ (response) by performing a randomized block experiment. The set of observed covariates is $\mathcal{C} := \mathcal{V}\backslash\{X, Y\}$.

Studying the causal effect of $X$ on $Y$ means learning the (interventional) probability distribution, $\mathbb{P}(Y|do(X))$. We can write $\mathbb{P}(Y = y|do(X = x))$ as

$$\mathbb{P}(Y = y|do(X = x)) = \frac{\mathbb{P}\left(Y = y \cap do(X = x)\right)}{\mathbb{P}\left(do(X = x)\right)} \tag{13}$$

For estimating the above probability, we perform an RBD by forming blocks using covariates in $\mathcal{Z} \subseteq \mathcal{V}\backslash\{X, Y\}$, and similar to the estimation of $\mathbb{E}\,(Y|do(X))$, we define an estimate of $\mathbb{P}(Y = y|do(X = x))$ as

$$\hat{\mathbb{P}}_{\mathcal{Z}}(Y = y|do(X = x)) := \sum_{\mathcal{Z}=z} \frac{\hat{\mathbb{P}}\left(Y = y \cap do(X = x) \cap \mathcal{Z} = z\right)}{\hat{\mathbb{P}}\left(do(X = x) \cap \mathcal{Z} = z\right)}\hat{\mathbb{P}}\left(\mathcal{Z} = z\right) \tag{14}$$

where $\hat{\mathbb{P}}\,(\cdot)$ are the sample relative frequencies.

It is desirable to select the $\mathcal{Z}$ such that $\mathrm{Var}\,(\hat{\mathbb{P}}_{\mathcal{Z}}(Y = y|do(X = x)))$ is minimized, i.e., maximum gain in precision. For ease of notations, we define $\mathrm{Var}\,(\hat{\mathbb{P}}_{\mathcal{Z}}(Y|do(X))) := \mathrm{Var}\,(\hat{\mathbb{P}}_{\mathcal{Z}}(Y = y|do(X = x)))$.

**Problem 1.** *Given a directed acyclic graph, $\mathcal{G}$, that depicts the causal relationships among some observed variables, $\mathcal{V}$, and unobserved variables, $\mathcal{U}$; treatment, $X \in \mathcal{V}$, and response, $Y \in \mathcal{V}$, obtain a smallest subset, $\mathcal{Z}^*$, of the set of observed covariates, $\mathcal{C} := \mathcal{V}\backslash\{X, Y\}$, such that*

$$\mathcal{Z}^* \in \arg\min_{\mathcal{Z}} \mathit{Var}\,(\hat{\mathbb{P}}_{\mathcal{Z}}(Y|do(X))). \tag{15}$$

**Note.** In Section 3.4, we discuss the idea of *stability* of the solution to Problem 1 where we talk about some covariates that should never be used while forming the blocks. We provide a method to obtain the set of such variables.

## 3 Methodology

In this section, we develop a methodology to find a solution to Problem 1. We first examine $\mathrm{Var}\,(\hat{\mathbb{P}}_{\mathcal{Z}}(Y|do(X)))$ as a function of $\mathcal{Z}$ to obviate some edges and nodes from the causal graph, $\mathcal{G}$. We next discuss some motivating examples and make key observations that lead to several lemmas for our main result. In the end, we provide our main result and develop an efficient algorithm for solving Problem 1.

### 3.1 Examining $\mathbf{Var}\,(\hat{\mathbb{P}}_{\mathcal{Z}}(Y|do(X)))$ as a function of $\mathcal{Z}$

In Problem 1, our interest is to minimize $\mathrm{Var}\,(\hat{\mathbb{P}}_{\mathcal{Z}}(Y|do(X)))$ as a function of $\mathcal{Z} \subseteq \mathcal{C}$. $\mathcal{Z}$ can affect $\mathrm{Var}\,(\hat{\mathbb{P}}_{\mathcal{Z}}(Y|do(X)))$ through the causal relationships from $\mathcal{Z}$ to $Y$, and from $\mathcal{Z}$ to $X$. Note that, $X$ is the treatment on which we are performing the intervention, i.e., we set the levels/values of $X$. This is equivalent to saying that $X$ is no longer affected by the rest of the variables. Therefore, $\mathcal{Z}$ also cannot affect $X$ when an intervention is performed on $X$. Therefore, $\mathcal{Z}$ can affect $\mathrm{Var}\,(\hat{\mathbb{P}}_{\mathcal{Z}}(Y|do(X)))$ only through the causal relationships from $\mathcal{Z}$ to $Y$. Therefore, a smallest subset of $\mathcal{C}$ that blocks all causal paths from $\mathcal{C}$ to $Y$ gives a solution to 1.

Denote the subgraph of $\mathcal{G}$ where all edges coming into $X$ were deleted as $\mathcal{G}_{\widetilde{X}}$.

**Lemma 1.** *For finding a solution to Problem 1, it is sufficient to work with the subgraph $\mathcal{G}_{\widetilde{X}}$.*

*Proof.* The proof follows from the fact that when an intervention is performed on $X$ then it is no longer affected by the rest of the variables. In terms of the causal graph, $\mathcal{G}$, it is the same as deleting all edges coming into $X$. □

**Definition 1.** $\mathcal{S}, \mathcal{T} \subseteq \mathcal{C}$ *are called equivalent sets (denoted as $\mathcal{S} \equiv \mathcal{T}$), if $Var(\hat{\mathbb{P}}_{\mathcal{S}}(Y|do(X))) = Var(\hat{\mathbb{P}}_{\mathcal{T}}(Y|do(X)))$.*

**Lemma 2.** *If $\mathcal{F}$ is the set of all covariates that do not have causal paths to $Y$, then $\mathcal{C}$ and $\mathcal{C}\backslash\mathcal{F}$ are equivalent sets.*

*Proof.* The proof follows from the fact that a set of covariates can affect $Var(\hat{\mathbb{P}}_{\mathcal{Z}}(Y|do(X)))$ only through the causal relationships from itself to $Y$. □

Following Lemma 1, we restrict ourselves to the subgraph $\mathcal{G}_{\widetilde{X}}$. Following Lemma 2, we further delete the covariates that do not have any causal paths to $Y$. Denote this new subgraph as $\mathcal{G}'_{\widetilde{X}}$ and $\mathcal{C}' = \mathcal{C}\backslash\mathcal{F}$. Therefore, a smallest subset of $\mathcal{C}$ that blocks all causal paths from $\mathcal{C}'$ to $Y$ in $\mathcal{G}'_{\widetilde{X}}$ gives a solution to 1.

### 3.2 Motivating Examples and Key Observations

All causal graphs in Figures 1 to 5 represent represent $\mathcal{G}'_{\widetilde{X}}$. Denote the set of parents of $W$ in $\mathcal{G}'_{\widetilde{X}}$ as $\mathcal{P}(W)$.

**Example 1.** For Figures 1 and 2, $\mathcal{P}(Y)\backslash\{X\} = \{V_1, V_2\}$ is sufficient to block all causal paths from $\mathcal{C}' = \{V_1, V_2, V_3, V_4\}$ to $Y$. Therefore, $\mathcal{Z}^* = \mathcal{P}(Y)\backslash\{X\}$.

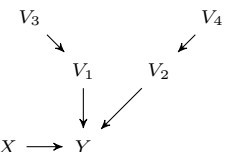

Figure 1: Example Markovian model (no latent structures).

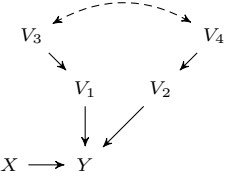

Figure 2: Example semi-Markovian model with latent variables involving only $\mathcal{C}'$.

**Example 2.** For Figures 3a and 3b, $\mathcal{P}(Y)\backslash\{X\} = \{V_1, V_2\}$ is not sufficient to block all causal paths from $\mathcal{C}' = \{V_1, V_2, V_3, V_4\}$ to $Y$. In Figure 3a, the reason is that blocking using $V_2$ opens the path $V_4 \to V_2 \leftarrow Y$ for $V_4$ to cause variability in $Y$. In Figure 3b, the reason is that the latent structure $Y \leftrightarrow V_4$ enables $V_4$ to cause $Y$ even when the path $V_4 \to V_2 \to Y$ is blocked. For both Figures 3a and 3b, $\mathcal{Z}^* = (\mathcal{P}(Y)\backslash\{X\}) \cup \{V_4\}$.

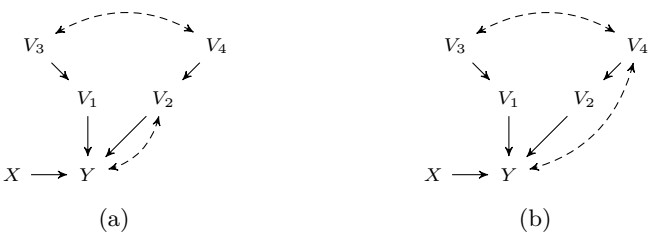

(a)                    (b)

Figure 3: Example semi-Markovian model with latent variables involving $\mathcal{C}'$ and $Y$.

**Example 3.** For Figure 4, despite $Y$ having descendants, $(\mathcal{P}(Y)\backslash\{X\}) \cup \{V_4\}$ blocks all causal paths from $\mathcal{C}'$ to $Y$. This is because descendants of $Y$ can never be a cause of variability in $Y$. In Figures 4a and 4b, $V_5$ (a descendant of $Y$) cannot cause $Y$ through the edge $Y \to V_5$. Moreover, $V_2$ (a parent of $Y$) cannot cause $Y$ through $V_5$ because $V_5$ is a collider on the path $V_2 \to V_5 \leftarrow Y$.

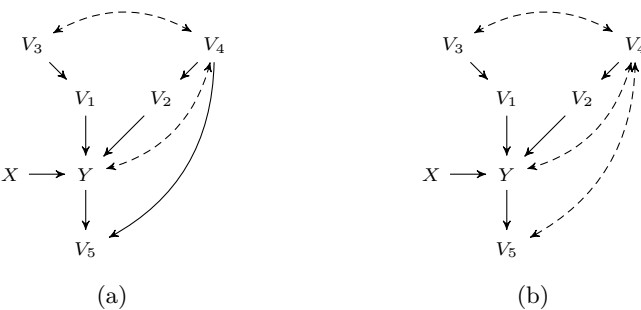

(a)          (b)

Figure 4: Example Semi-Markovian Models with latent variables involving $\mathcal{C}'$ and $Y$, and $Y$ having descendants.

Denote the set of ancestors (including itself) of $W$ in $\mathcal{G}'_{\widetilde{X}}$ as $\mathcal{A}(W)$. Denote the subgraph of $\mathcal{G}'_{\widetilde{X}}$ restricted to $\mathcal{A}(Y)$ as $\mathcal{G}'_{\widetilde{X},\mathcal{A}(Y)}$. Based on the insights from Examples 1 to 3, we write the following lemma.

**Lemma 3.** *For finding a solution to Problem 1, it is sufficient to work with the subgraph $\mathcal{G}'_{\widetilde{X},\mathcal{A}(Y)}$.*

*Proof.* The proof follows from the fact that the descendants of a node can never be its cause even in the presence of latent structures. □

### 3.3 Solution to Problem 1

We now combine the insights from Lemmas 1 to 3 to provide a solution to Problem 1.

**Theorem 1.** *The smallest $\mathcal{Z}$ such that $Y \perp (\mathcal{A}(Y)\backslash\{X,Y\})\backslash\mathcal{Z}|\mathcal{Z}$ in $\mathcal{G}'_{\widetilde{X},\mathcal{A}(Y)}$ is a solution to Problem 1.*

*Proof.* The proof follows from the fact that $\mathcal{Z}$ such that $Y \perp (\mathcal{A}(Y)\backslash\{X,Y\})\backslash\mathcal{Z}|\mathcal{Z}$ in $\mathcal{G}'_{\widetilde{X},\mathcal{A}(Y)}$ blocks all causal paths from $\mathcal{C}'$ to $Y$. □

Theorem 1 provides a sufficient condition for a set $\mathcal{Z}$ to be a solution to Problem 1. We next provide a method to construct a set $\mathcal{Z}$ that satisfies this sufficient condition.

Let a path composed entirely of bi-directed edges be called a *bi-directed path*.

**Definition 2.** *(Tian & Pearl, 2002). For a graph $\mathcal{G}$, the set of observed variables $\mathcal{V}$, can be partitioned into disjoint groups by assigning two variables to the same group if and only if they are connected by a bi-directed path. Assume that $V$ is therefore partitioned into $k$ groups $\mathcal{S}_1,\ldots,\mathcal{S}_k$. We will call each $\mathcal{S}_j; j = 1,\ldots,k$, a c-component of $V$ in $\mathcal{G}$ or a c-component (abbreviating confounded component) of $\mathcal{G}$. The $\mathcal{S}_j$ such that $W \in S_j$ is called the c-component of $W$ in $\mathcal{G}$ and is denoted as $C_{W,\mathcal{G}}$. As $\{\mathcal{S}_1,\ldots,\mathcal{S}_k\}$ is a partition of $\mathcal{V}$, c-component of a variable always exists and is unique.*

In Example 2, we observed that due to the presence of latent structures involving $Y$, $\mathcal{P}(Y)$ was not sufficient to block all causal paths from $\mathcal{C}'$ to $Y$. However the parents of c-component of $Y$ in $\mathcal{G}'_{\widetilde{X},\mathcal{A}(Y)}$ would be sufficient for that purpose.

**Theorem 2.** *The smallest $\mathcal{Z}$ such that $Y \perp (\mathcal{A}(Y)\backslash\{X,Y\})\backslash\mathcal{Z}|\mathcal{Z}$ is the set of parents of c-component of $Y$ in $\mathcal{G}'_{\widetilde{X},\mathcal{A}(Y)}$ excluding $X$ denoted as $\mathcal{P}(C_{Y,\mathcal{G}_{\widetilde{X},\mathcal{A}(Y)}})\backslash\{X\}$.*

*Proof.* The proof follows from Corollary 1 (Tian, 2002), which states that a node is independent of its ancestors excluding the parents of its c-component given the parents of its c-component. □

**Note.** In a Markovian model there are no bi-directed paths, Therefore, the elementary partition of $\mathcal{A}(Y)$ is the c-component of $\mathcal{G}'_{\widetilde{X},\mathcal{A}(Y)}$. Therefore, $C_Y = \{Y\}$, and $\mathcal{P}(C_{Y,\mathcal{G}_{\widetilde{X},\mathcal{A}(Y)}})\backslash\{X\} = \mathcal{P}(Y)\backslash\{X\}$ in $\mathcal{G}'_{\widetilde{X},\mathcal{A}(Y)}$. This matches with our insight in Example 1, where we saw that $Y \perp (\mathcal{A}(Y)\backslash\{X,Y\})\backslash(\mathcal{P}(Y)\backslash\{X\})|(\mathcal{P}(Y)\backslash\{X\})$.

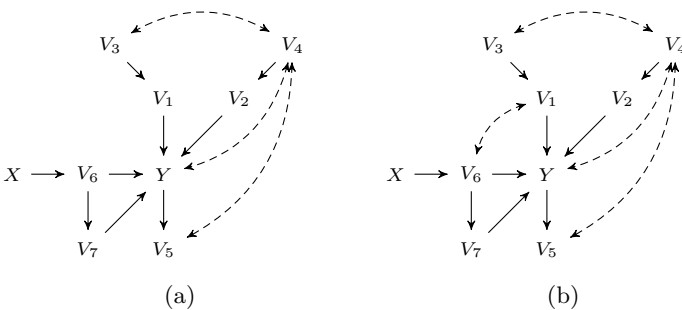

Figure 5: Example semi-Markovian models with post-treatment ancestors of $Y$.

## 3.4 Stability of the Solution to Problem 1

We next discuss some concerns with using $\mathcal{Z}^*$ (a solution to Problem 1) as a blocking set. We provide a method to address those concerns.

**Example 4.** In Figure 5a and Figure 5b, there are some covariates which are ancestors of $Y$ and descendants of $X$. Such covariates are called post-treatment ancestors (excluding itself) of the response. Blocks created using these covariates are not well-defined. For instance, we are interested in studying the causal effect of a drug on blood pressure. Suppose anxiety level mediates the effect of drug on blood pressure, i.e., drug $\rightarrow$ anxiety $\rightarrow$ blood pressure. If we create blocks using anxiety then the blocks will change (i.e., become unstable) during the experiment because change in the drug level will cause the anxiety level to change. Therefore, it is reasonable not to create blocks using these covariates. With this insight, for Figure 5a, $(\mathcal{P}(Y)\backslash\{X\}) \cup \{V_4\}$ blocks all (pre-treatment) causal paths from $\mathcal{C}'$ to $Y$. However, we cannot simply delete these covariates from the subgraph $\mathcal{G}'_{\widetilde{X},\mathcal{A}(Y)}$ of interest. This is because presence of latent structures between pre- and post-treatment ancestors of the response affects blocking set of interest. For instance, in Figure 5b, $V_2$ is a collidor on the path $V_3 \rightarrow V_1 \leftarrow V_6$ keeping $V_1$ in the blocking set will open this path for $V_3$ to cause variability in $Y$. Therefore, for Figure 5a, we need $(\mathcal{P}(Y)\backslash\{X\}) \cup \{V_3, V_4\}$ to block all (pre-treatment) causal paths from $\mathcal{C}'$ to $Y$.

Denote the post-treatment ancestors (excluding itself) of the response while studying the causal effect $Y|do(X)$ as $\mathcal{M}(Y|do(x))$.

**Definition 3.** *For studying the causal effect of $X$ on $Y$, a blocking set $\mathcal{Z}$, is said to be stable if $\mathcal{Z} \cap \mathcal{M}(Y|do(x)) = \phi$.*

Denote the set of descendants (including itself) of $W$ in $\mathcal{G}'_{\widetilde{X},\mathcal{A}(Y)}$ as $\mathcal{D}(\cdot)$.

**Lemma 4.** $\mathcal{M}(Y|do(x)) = (\mathcal{D}(X) \cap \mathcal{A}(Y))\backslash\{X, Y\}$.

*Proof.* The proof follows from the definition of post-treatment ancestors (excluding itself) of $Y$. $\qquad\square$

**Theorem 3.** *A stable solution to Problem 1 is $\mathcal{Z}^* = (\mathcal{P}(C_{Y,\mathcal{G}'_{\widetilde{X},\mathcal{A}(Y)}}\backslash\{X\})\backslash\mathcal{M}(Y|do(x))$.*

*Proof.* The proof follows from Lemma 4. $\qquad\square$

## 3.5 Final Algorithm

We next provide steps to obtain a stable solution to Problem 1 based on Theorems 1 to 3. First, we reduce $\mathcal{G}$ to $\mathcal{G}'_{\widetilde{X},\mathcal{A}(Y)}$. Next, we obtain the set $\mathcal{Z} = \mathcal{P}(C_{Y,\mathcal{G}'_{\widetilde{X},\mathcal{A}(Y)}})\backslash\{X\}$ which is the smallest set such that $Y \perp (\mathcal{A}(Y)\backslash\{X, Y\})\backslash\mathcal{Z}|\mathcal{Z}$ in $\mathcal{G}'_{\widetilde{X},\mathcal{A}(Y)}$. Finally, we drop $\mathcal{M}(Y|do(x))$ from $\mathcal{Z}$ to get a stable solution to Problem 1.

Denote the edge $W_1 \to W_2$ as *directed-edge*$(W_1, W_2)$, a path consisting of directed edges from $W_1$ to $W_2$ as *directed-path*$(W_1, W_2)$, and a path consisting of all bi-directed edges from $W_1$ to $W_2$ as *bi-directed-path*$(W_1, W_2)$. Algorithm 1 outlines the psuedocode of the proposed algorithm to obtain a stable solution to Problem 1.

---

**Algorithm 1** Stable-Causal-Blocking $(\mathcal{G}, X, Y)$

---

1: **for** $i$ in $1, \ldots, |\mathcal{V}|$ **do** $\qquad\qquad\qquad\qquad\qquad\qquad\qquad\qquad$ ▷ Reducing $\mathcal{G}$ to $\mathcal{G}_{\widetilde{X}}$
2: $\quad$ **if** $\exists$ *directed-edge*$(V_i, X)$ **then**
3: $\qquad$ delete *directed-edge*$(V_i, X)$ from $\mathcal{G}$
4: **for** $i$ in $1, \ldots, |\mathcal{V}|$ **do** $\qquad\qquad\qquad\qquad\qquad\qquad\qquad\qquad$ ▷ Reducing $\mathcal{G}_{\widetilde{X}}$ to $\mathcal{G}'_{\widetilde{X}}$
5: $\quad$ **if** $\nexists$ *directed-path*$(V_i, Y)$ **then**
6: $\qquad$ delete $V_i$ from $\mathcal{G}$
7: $\mathcal{A}'(Y) \leftarrow \phi$
8: **for** $i$ in $1, \ldots, |\mathcal{V}|$ **do** $\qquad\qquad\qquad\qquad\qquad\qquad$ ▷ Reducing $\mathcal{G}'_{\widetilde{X}}$ to $\mathcal{G}'_{\widetilde{X}, \mathcal{A}(Y)}$
9: $\quad$ **if** $\nexists$ path$(V_i, Y)$ **then**
10: $\qquad$ delete $V_i$ from $\mathcal{G}$; $\mathcal{A}'(Y) \leftarrow \mathcal{A}'(Y) \cup \{V_i\}$
11: $\mathcal{A}(Y) = \mathcal{V} \backslash \mathcal{A}'(Y)$
12: $\mathcal{S}_i = \{V_i\}, i = 1, \ldots, |\mathcal{V}|$ $\qquad\qquad\qquad\qquad\qquad\qquad\qquad$ ▷ Finding $C_{Y, \mathcal{G}_{\widetilde{X}, \mathcal{A}(Y)}}$
13: **for** $i$ in $1, \ldots, |\mathcal{V}|$ **do**
14: $\quad$ **for** $j$ in $1, \ldots, i$ **do**
15: $\qquad$ **if** $\exists$ *bi-directed-path*$(V_i, V_j)$ **then**
16: $\qquad\qquad$ $\mathcal{S}_i \leftarrow \mathcal{S}_i \cup \mathcal{S}_j$; $\mathcal{S}_j \leftarrow \phi$
17: $i \leftarrow 1$; $\mathcal{S} \leftarrow \mathcal{S}_1$
18: **while** $Y \notin \mathcal{S}$ **do**
19: $\quad$ $i \leftarrow i + 1$; $\mathcal{S} \leftarrow \mathcal{S}_i$
20: $\mathcal{P}(\mathcal{S}) = \phi$ $\qquad\qquad\qquad\qquad\qquad\qquad\qquad\qquad$ ▷ Finding $\mathcal{P}(C_{Y, \mathcal{G}_{\widetilde{X}, \mathcal{A}(Y)}})$
21: **for** $i$ in $1, \ldots, |\mathcal{V}|$ **do**
22: $\quad$ **for** $W$ in $\mathcal{S}$ **do**
23: $\qquad$ **if** $\exists$ *directed-edge*$(V_i, W)$ **then**
24: $\qquad\qquad$ $\mathcal{P}(\mathcal{S}) \leftarrow \mathcal{P}(\mathcal{S}) \cup \{V_i\}$
25: $\mathcal{D}(X) \leftarrow \phi$ $\qquad\qquad\qquad\qquad\qquad\qquad\qquad\qquad\qquad\quad$ ▷ Finding $\mathcal{D}(X)$
26: **for** $i$ in $1, \ldots, |\mathcal{V}|$ **do**
27: $\quad$ **if** $\nexists$ *directed-path*$(X, V_i)$ **then**
28: $\qquad$ $\mathcal{D}(X) \leftarrow \mathcal{D}(X) \cup \{V_i\}$
29: $\mathcal{Z} \leftarrow (\mathcal{P}(\mathcal{S}) \backslash \{X\}) \backslash (\mathcal{D}(X) \cap \mathcal{A}(Y)) \backslash \{X, Y\}$.
30: **return** $\mathcal{Z}$

---

### 3.6 Time Complexity of Algorithm 1

We next show that Algorithm 1 is an efficient polynomial-time algorithm. Finding if there exists a path between any two nodes in a directed graph can be done using breath-first-search which has the worst-case time complexity of $O(|\mathcal{V}|^2)$. Algorithm 1 first reduces $\mathcal{G}$ to $\mathcal{G}'_{\widetilde{X}, \mathcal{A}(Y)}$ which involves

1. finding variables with edges going into $X$: $O(|\mathcal{V}|)$,

2. finding variables with no path into $Y$: $O(|\mathcal{V}|^3)$, and

3. finding all ancestors of $Y$: $O(|\mathcal{V}|^3)$.

Algorithm 1 next finds the *c*-component of $Y$ involves traversing all pairs of variables: $O(|\mathcal{V}|^2)$, and finding if there exists a bi-directed path between any pair of variables: $O(|\mathcal{V}|^2)$. Therefore, the total time at this step is $O(|\mathcal{V}|^4)$. Finally Algorithm 1 finds $\mathcal{M}(Y|do(x))$ which involves

1. finding descendants of $X$: $O(|\mathcal{V}|^3)$, and

2. finding ancestors of $Y$: $O(|\mathcal{V}|^3)$.

Therefore, the time complexity of Algorithm 1 is $O(|\mathcal{V}|^4)$.

### 3.7 Implementing Algorithm 1

We demonstrate the implementation of Algorithm 1 to obtain a stable solution to Problem 1 for a general semi-Markovian causal graph, $\mathcal{G}$ given in Figure 6.

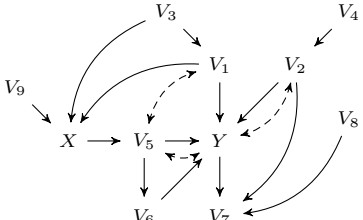

Figure 6: The Original Graph $\mathcal{G}$.

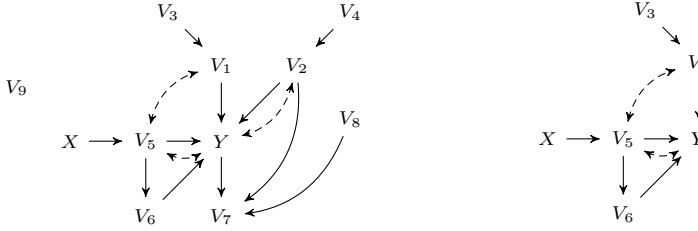

Figure 7: The Graph $\mathcal{G}_{\widetilde{X}}$.
Figure 8: The Graph $\mathcal{G}_{\widetilde{X},\mathcal{A}(Y)}$.

We first obtain the graph $G_{An_{\widetilde{X}}(Y)}$ as displayed in Figure 8. We next obtain the $c$-components (following Definition 2) of $\mathcal{G}_{\widetilde{X},\mathcal{A}(Y)}$ are $\{X\}, \{V_1, V_5, Y, V_2\}, \{V_3\}, \{V_4\}$, and the $c$-component of $Y$ is $\{V_1, V_5, Y, V_2\}$. We next obtain the set of parents of the $c$-component of $Y$ as $\{X, V_1, V_2, V_3, V_4, V_5\}$. The set $\mathcal{D}(X) \cap \mathcal{A}(Y)$ is $\{X, Y, V_5, V_6\}$. Finally, we obtain the final set as $(\{X, V_1, V_2, V_3, V_4, V_5\}\backslash\{X\})\backslash(\{X, Y, V_5, V_6\}\backslash\{X, Y\}) = \{V_1, V_2, V_3, V_4\}$.

## 4 Conclusion and Future Work

We investigated the problem of finding a stable set of covariates to be used for forming blocks, that minimizes the variance of the causal effect estimates. By leveraging the underlying causal graph, we provided an efficient algorithm to obtain such a set for a general semi-Markovian causal model. In the future, we are interested in finding stable solutions to Problem 1 with additional cardinality or knapsack constraint on the feasible solutions.

# A   Appendix

## A.1   Expectation and Variance of $\hat{\beta}$

We calculate the expectation of $\hat{\beta}$ as

$$\mathbb{E}\left(\hat{\beta}\right) = \mathbb{E}\left(\bar{Y}(1)\right) - \mathbb{E}\left(\bar{Y}(0)\right). \tag{16}$$

In general, due to non-homogeneous experimental units, $\mathbb{E}\left(\bar{Y}(x)\right) \neq \mathbb{E}\left(Y(x)\right), x = 0, 1$. Let $\mathcal{Z}$ be the covariates causing non-homogeneity, rewrite equation 16 as

$$\mathbb{E}\left(\hat{\beta}\right) = \mathbb{E}_{\mathcal{Z}}\left(\mathbb{E}\left(\bar{Y}(1)|\mathcal{Z}\right)\right) - \mathbb{E}_{\mathcal{Z}}\left(\mathbb{E}\left(\bar{Y}(0)|\mathcal{Z}\right)\right). \tag{17}$$

For fixed covariates the experimental units are identical. Therefore, $\mathbb{E}\left(\bar{Y}(x)|\mathcal{Z}\right) = \mathbb{E}\left(Y(1)|\mathcal{Z}\right), x = 0, 1$. Thus,

$$\mathbb{E}\left(\hat{\beta}\right) = \mathbb{E}_{\mathcal{Z}}\left(\mathbb{E}\left(Y(x)|\mathcal{Z}\right)\right) - \mathbb{E}_{\mathcal{Z}}\left(\mathbb{E}\left(Y(0)|\mathcal{Z}\right)\right), \tag{18}$$

$$= \mathbb{E}\left(Y(1)\right) - \mathbb{E}\left(Y(0)\right) = \beta. \tag{19}$$

We next calculate the variance of $\hat{\beta}$ as

$$\mathrm{Var}\left(\hat{\beta}\right) = \mathrm{Var}\left(\bar{Y}(1) - \bar{Y}(0)\right). \tag{20}$$

Let $\mathcal{Z}$ be the covariates causing non-homogeneity, we rewrite equation 20 as

$$\mathrm{Var}\left(\hat{\beta}\right) = \mathrm{Var}_{\mathcal{Z}}\left(\mathbb{E}\left((\bar{Y}(1) - \bar{Y}(0))|\mathcal{Z}\right)\right) + \mathbb{E}_{\mathcal{Z}}\left(\mathrm{Var}\left((\bar{Y}(1) - \bar{Y}(0))|\mathcal{Z}\right)\right), \tag{21}$$

$$= \mathrm{Var}_{\mathcal{Z}}\left(\mathbb{E}\left(\bar{Y}(1)|\mathcal{Z}\right) - \mathbb{E}\left(\bar{Y}(0)|\mathcal{Z}\right)\right) + \mathbb{E}_{\mathcal{Z}}\left(\mathrm{Var}\left(\bar{Y}(1)|\mathcal{Z}\right) + \mathrm{Var}\left(\bar{Y}(0)|\mathcal{Z}\right)\right). \tag{22}$$

equation 22 uses linearity of expectation and randomization.

For fixed covariates the experimental units are identical. Therefore, $\mathbb{E}\left(\bar{Y}(x)|\mathcal{Z}\right) = Y(x)|\mathcal{Z}, x = 0, 1$, and $\mathrm{Var}\left(\bar{Y}(x)|\mathcal{Z}\right) = \frac{1}{n_{x,z}}\mathrm{Var}\left(Y(x)|\mathcal{Z}\right), x = 0, 1$. Therefore,

$$\mathrm{Var}\left(\hat{\beta}\right) = \mathrm{Var}_{\mathcal{Z}}\left(\mathbb{E}\left(Y(1)|\mathcal{Z}\right) - \mathbb{E}\left(Y(0)|\mathcal{Z}\right)\right) + \mathbb{E}_{\mathcal{Z}}\left(\frac{\mathrm{Var}\left(Y(1)|\mathcal{Z}\right)}{n_{1,z}} + \frac{\mathrm{Var}\left(Y(0)|\mathcal{Z}\right)}{n_{0,z}}\right) \tag{23}$$

Define $\beta(\mathcal{Z}) := \mathbb{E}\left(Y(1)|\mathcal{Z}\right) - \mathbb{E}\left(Y(0)|\mathcal{Z}\right)$. Therefore,

$$\mathrm{Var}\left(\hat{\beta}\right) = \mathrm{Var}_{\mathcal{Z}}\left(\beta(\mathcal{Z})\right) + \mathbb{E}_{\mathcal{Z}}\left(\frac{\mathrm{Var}\left(Y(1)|\mathcal{Z}\right)}{n_{1,z}} + \frac{\mathrm{Var}\left(Y(0)|\mathcal{Z}\right)}{n_{0,z}}\right) \tag{24}$$

$$= \mathbb{E}_{\mathcal{Z}}\left(\beta(\mathcal{Z}) - \mathbb{E}_{\mathcal{Z}}(\beta(\mathcal{Z}))\right)^2 + \mathbb{E}_{\mathcal{Z}}\left(\frac{\mathrm{Var}\left(Y(1)|\mathcal{Z}\right)}{n_{1,z}} + \frac{\mathrm{Var}\left(Y(0)|\mathcal{Z}\right)}{n_{0,z}})\right) \tag{25}$$

$$= \mathbb{E}_{\mathcal{Z}}\left(\beta(\mathcal{Z}) - \beta)\right)^2 + \mathbb{E}_{\mathcal{Z}}\left(\frac{\mathrm{Var}\left(Y(1)|\mathcal{Z}\right)}{n_{1,z}} + \frac{\mathrm{Var}\left(Y(0)|\mathcal{Z}\right)}{n_{0,z}}\right) \tag{26}$$

equation 26 uses $\mathbb{E}_{\mathcal{Z}}(\beta(\mathcal{Z})) = \mathbb{E}_{\mathcal{Z}}(\mathbb{E}\left(Y(1)|\mathcal{Z}\right)) - \mathbb{E}_{\mathcal{Z}}(\mathbb{E}\left(Y(0)|\mathcal{Z}\right)) = \mathbb{E}\left(Y(1)\right) - \mathbb{E}\left(Y(0)\right) = \beta$.

## A.2   Expectation and Variance of $\hat{\beta}_{\mathcal{Z}}$

We calculate the expectation of $\hat{\beta}_{\mathcal{Z}}$ as

$$\mathbb{E}\left(\hat{\beta}_{\mathcal{Z}}\right) = \mathbb{E}\left(\sum_{\mathcal{Z}}\hat{\mathbb{P}}(\mathcal{Z})\left(\bar{Y}(1)|\mathcal{Z} - \bar{Y}(0)|\mathcal{Z}\right)\right), \tag{27}$$

$$= \sum_{\mathcal{Z}}\mathbb{E}\left(\hat{\mathbb{P}}(\mathcal{Z})(\bar{Y}(1)|\mathcal{Z} - \bar{Y}(0)|\mathcal{Z})\right), \tag{28}$$

$$= \sum_{\mathcal{Z}}\mathbb{E}\left(\hat{\mathbb{P}}(\mathcal{Z})\right)\left(\mathbb{E}\left(\bar{Y}(1)|\mathcal{Z}\right) - \mathbb{E}\left(\bar{Y}(0)|\mathcal{Z}\right)\right), \tag{29}$$

$$= \sum_{\mathcal{Z}}\mathbb{P}(\mathcal{Z})\left(\mathbb{E}\left(\bar{Y}(1)|\mathcal{Z}\right) - \mathbb{E}\left(\bar{Y}(0)|\mathcal{Z}\right)\right), \tag{30}$$

$$= \mathbb{E}_{\mathcal{Z}}\left(\mathbb{E}\left(\bar{Y}(1)|\mathcal{Z}\right) - \mathbb{E}\left(\bar{Y}(0)|\mathcal{Z}\right)\right) = \beta. \tag{31}$$

We next calculate the variance of $\beta_{\mathcal{Z}}$ as

$$\text{Var}\,(\hat{\beta}_{\mathcal{Z}}) = \text{Var}\,_{\mathcal{Z}}\left(\mathbb{E}\left(\sum_{\mathcal{Z}}(\bar{Y(1)}|\mathcal{Z} - \bar{Y(0)}|\mathcal{Z})\hat{\mathbb{P}}\,(Z)\right)\right) + \mathbb{E}\,_{\mathcal{Z}}\left(\text{Var}\left(\sum_{\mathcal{Z}}(\bar{Y(1)}|\mathcal{Z} - \bar{Y(0)}|\mathcal{Z})\hat{\mathbb{P}}\,(Z)\right)\right),$$
(32)

$$= \text{Var}\,_{\mathcal{Z}}\left(\sum_{\mathcal{Z}}\mathbb{E}\,(\bar{Y(1)}|\mathcal{Z} - \bar{Y(0)}|\mathcal{Z})\hat{\mathbb{P}}\,(Z)\right) + \mathbb{E}\,_{\mathcal{Z}}\left(\sum_{\mathcal{Z}}\text{Var}\,(\bar{Y(1)}|\mathcal{Z} - \bar{Y(0)}|\mathcal{Z})\hat{\mathbb{P}}\,(Z)\right),$$
(33)

$$= \text{Var}\,_{\mathcal{Z}}\left(\sum_{\mathcal{Z}}\mathbb{E}\,(\bar{Y}(1)|\mathcal{Z}) - \mathbb{E}\,(\bar{Y}(0)|\mathcal{Z})\hat{\mathbb{P}}\,(Z)\right) + \mathbb{E}\,_{\mathcal{Z}}\left(\sum_{\mathcal{Z}}\left(\text{Var}\,(\bar{Y}_1|\mathcal{Z}) + \text{Var}\,(\bar{Y}_0|\mathcal{Z})\right)\hat{\mathbb{P}}\,(Z)\right),$$
(34)

equation 33 and equation 34 use linearity of expectation and randomization.

For fixed covariates the experimental units are identical. Therefore, $\mathbb{E}\,(\bar{Y}(x)|\mathcal{Z}) = \mathbb{E}\,(Y(x)|\mathcal{Z}), x = 0, 1$, and $\text{Var}\,(\bar{Y}(x)|\mathcal{Z}) = \frac{1}{n_{x,z}}\text{Var}\,(Y(x)|\mathcal{Z}), x = 0, 1$. Therefore,

$$\text{Var}\,(\hat{\beta}_{\mathcal{Z}}) = \text{Var}\,_{\mathcal{Z}}\left(\sum_{\mathcal{Z}}\beta(Z)\hat{\mathbb{P}}(\mathcal{Z})\right) + \mathbb{E}\,_{\mathcal{Z}}\left(\sum_{\mathcal{Z}}\left(\frac{\text{Var}\,(Y(1)|\mathcal{Z})}{n_{1,z}} + \frac{\text{Var}\,(Y(0)|\mathcal{Z})}{n_{0,z}}\right)\hat{\mathbb{P}}(\mathcal{Z})\right),$$
(35)

$$= \mathbb{E}\,_{\mathcal{Z}}\left(\sum_{\mathcal{Z}}\beta(Z)\hat{\mathbb{P}}(\mathcal{Z}) - \mathbb{E}\,_{\mathcal{Z}}\left(\sum_{\mathcal{Z}}\beta(Z)\hat{\mathbb{P}}(\mathcal{Z})\right)\right)^2 + \sum_{\mathcal{Z}}\frac{\text{Var}\,(Y(1)|\mathcal{Z})}{n_{1,z}}\mathbb{E}\,_{\mathcal{Z}}(\hat{\mathbb{P}}(\mathcal{Z}))$$

$$+ \sum_{\mathcal{Z}}\frac{\text{Var}\,(Y(0)|\mathcal{Z})}{n_{0,z}}\mathbb{E}\,_{\mathcal{Z}}(\hat{\mathbb{P}}(\mathcal{Z})),$$
(36)

$$= \mathbb{E}\,_{\mathcal{Z}}\left(\sum_{\mathcal{Z}}\beta(Z)\hat{\mathbb{P}}(\mathcal{Z}) - \beta\right)^2 + \sum_{\mathcal{Z}}\frac{\text{Var}\,(Y(1)|\mathcal{Z})}{n_{1,z}}\mathbb{P}(Z) + \sum_{\mathcal{Z}}\frac{\text{Var}\,(Y(0)|\mathcal{Z})}{n_{0,z}}\mathbb{P}(Z),$$
(37)

$$= \mathbb{E}\,_{\mathcal{Z}}\left(\sum_{\mathcal{Z}}\beta(Z)\hat{\mathbb{P}}(\mathcal{Z}) - \beta\right)^2 + \mathbb{E}\,_{\mathcal{Z}}\left(\frac{\text{Var}\,(Y(1)|\mathcal{Z})}{n_{1,z}}\right) + \mathbb{E}\,_{\mathcal{Z}}\left(\frac{\text{Var}\,(Y(1)|\mathcal{Z})}{n_{1,z}}\right),$$
(38)

$$= \mathbb{E}\,_{\mathcal{Z}}\left(\frac{\text{Var}\,(Y(1)|\mathcal{Z})}{n_{1,z}} + \frac{\text{Var}\,(Y(0)|\mathcal{Z})}{n_{0,z}}\right) + \mathbb{E}\,_{\mathcal{Z}}\left(\sum_{\mathcal{Z}}\beta(Z)\hat{\mathbb{P}}(\mathcal{Z}) - \beta\right)^2.$$
(39)

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
