# OpenReview forum: "Leveraging Causal Graphs for Blocking in Randomized Experiments"
_TMLR — Rejected by TMLR_

### Review · Reviewer_YDuY · 2022-06-08

**Summary Of Contributions:**

The claimed contribution is given on p2 in the two sentences which constitute Section 1.3: to (provide an algorithm to) find "stable" sets of covariates to form blocks (given a causal graph and a treatment whose causal effect on an outcome we seek to estimate). Note that "stable" is not defined until p8, 6 pages after it is used. The basic goal is to reduce the variance in estimation of causal effects. The algorithm is duly given in Section 3.5 and its time complexity analysed.

**Broader Impact Concerns:**

I don't have any concerns in this respect.

**Requested Changes:**

I would suggest the authors familiarise themselves further with existing work on estimating causal effects (using given causal models). This might suggest new work that needs doing (improvements on existing methods, problems that remain unsolved). I suspect a paper containing such new work would be radically different from the current submission and so there is no list of "adjustments" which would render the current paper acceptable.

Here are some typos etc:

The authors use the phrase "precisely estimate the causal effects" to
mean "estimate the causal effects with minimized variance". I would
advise avoiding the phrase "precisely estimate" since it literally means
to find an estimate which is always the true value.

For demonstrating the gain -> To demonstrate the gain

of the that -> of the

such that units with each stratum -> such that units within each stratum

For ease of notations -> For ease of notation

collidor -> collider

**Strengths And Weaknesses:**

The paper addresses an important problem and provides some interesting references. Beyond this, it is unfortunately hard to find strengths.
The key problem is that this paper ignores highly relevant existing work.

The paper contains the following paragraph:
"Cinelli et al. (2020) provides a good discussion on good and bad
controls for different causal graphs. However, they do not provide a
general algorithm to address the problem for a general causal
graph. In contrast, we provide an efficient algorithm to obtain the
set of covariates to be used for forming blocks for a general
semi-Markovian causal model."

But in Cinelli et al. (2020) there are references to software packages
which do provide a "general algorithm". As Cinelli et al state: " In
other words, given a causal diagram, the problem of deciding which
variables are good or bad controls has been automatized."

Since "forming blocks" and "deciding which variables are good or bad
controls" is the same task it follows that the software packages cited
by Cinelli et al already provide algorithms (indeed implemented
software) to form blocks from a given causal model. For example, the
dagitty manual states that in dagitty: "Functions include
identification of minimal sufficient adjustment sets for estimating
causal effects"

Here are some other problems:

(1) I see no need to eg include a proof for the unbiasedness of
\hat{\beta}, one could just point to the literature. For example, we
have this from "Understanding and misunderstanding randomized controlled
trials" by Deaton and Cartwright. (Soc Sci Med. 2018 August ; 210: 2–21.):
"The estimate of the average treatment effect is simply the difference
between the means in the two groups, and it has a standard error that
can be estimated using the statistical theory that applies to the
difference of two means, on which more below. The difference in means
is an unbiased estimator of the mean treatment effect."

(2) What does "If \hat{P}(Z) is as good as P(Z)" mean? The former is an
estimate of the latter. Does it mean that \hat{P}(Z) = P(Z)? This
would be a highly optimistic assumption to make, but this is
what the following text implies.

(3) Although the authors never explicitly state it, by Section 2.2 it
becomes clearer that they are implicitly restricting their paper to
*discrete* (or discretised) covariates. This is a very big
restriction.

(4) Equation (13) (and (14)) shows a misunderstanding of what an
interventional distribution is. For the RHS of this equation to make
sense "do(X=x)" would have to be an event co-existing with the event
"Y=y" in some sigma-algebra. But an interventional distribution is not a
regular conditional distribution.

(5) p2: We have the text "observed/endogenous" and
"unobserved/exogenous". I do not understand this text. Endogenous
variables can be observed or unobserved. Similarly for exogenous. Does
this text mean the authors only consider models where all observed
variables are endogenous and all unobserved variables are exogenous?

---

### Review · Reviewer_R2YL · 2022-06-10

**Summary Of Contributions:**

This paper concerns itself for the selection of a set of variables which should be accounted for when designing an experiment using blocking. The authors assume the existence of a known causal graph and then describe an identification criterion for finding the sufficient set of variables to include in the design such that variance is minimized. The authors propose to begin with the mutiliated graph, removing all incoming edges from the treatment (denoted X) and then applying an algorithm which seeks to find the smallest set of variables affecting Y which are not post-treatment for adjustment.

**Broader Impact Concerns:**

I don't see any immediate concerns here.

**Requested Changes:**

It would be great if the authors could incorporate some of the above comments listed in weaknesses. I would also like to see a larger contextualization of the work, both within the surrounding literature and within the task area itself. The authors frame this entire paper around blocking but as I mention above there are many more methods people use. Further, there needs to be a discussion of the no free lunch theorem that accompanies experimental design. It likely also makes sense to include regression adjustment, as there doesn't seem to be anything within the work that precludes the application to that setting. I'd also like to see a set of simulations to demonstrate the relative efficacy of the proposed approach.

**Strengths And Weaknesses:**

Strengths:
* I think this is an interesting task definition, striving to combine the problems of identification and experimental design.
* The authors write very clearly and do a nice job of describing their work.

Weaknesses:
* Why is this framed in terms of blocking? Is there something special about blocking here that makes this algorithm particularly applicable? It's not clear to me that it is, and it would seem that the authors are endeavoring to make a more general paper about choosing variables to include in an experimental design.

* The problem statement requires assumptions not made plain within the work. The authors seem to claim that blocking will always improve the precision of the estimate. However this is untrue. There are functions for which incorporating covariates into design will result in _worse_ variance than complete randomization. See Kallus below for a simulation.

* This work does not engage with the modern literature on experimental design in a meaningful way. The authors' cite some classic papers on blocking and the work of Moore and Higgins, but miss a number of very relevant works which I think materially affects the substance of the paper. In particular:
   (1) Rotnitzky, Andrea, and Ezequiel Smucler. "Efficient Adjustment Sets for Population Average Causal Treatment Effect Estimation in Graphical Models." J. Mach. Learn. Res. 21 (2020): 188-1. It's not entirely clear to me why this work would not be best described as a subcase of what is described by Rotnitzky & Smucler.
   (2) Kallus, Nathan. "Optimal a priori balance in the design of controlled experiments." Journal of the Royal Statistical Society: Series B (Statistical Methodology) 80.1 (2018): 85-112. This paper discusses the problem of experimental design in terms of variance reduction and describes the functional assumptions placed on the potential outcome by blocking.
  (3) Harshaw, Christopher, et al. "Balancing covariates in randomized experiments with the Gram--Schmidt Walk design." arXiv preprint arXiv:1911.03071 (2019). Shows that discrepancy minimization (difference in means between treatment and control covariate groups) implicitly performs ridge regression.
 (4) Li, Xinran, and Peng Ding. "Rerandomization and regression adjustment." Journal of the Royal Statistical Society: Series B (Statistical Methodology) 82.1 (2020): 241-268.

* The motivation to consider the mutilated graph is not well motivated to me. I've only experienced people advocating for variables that would otherwise be confounders in the observed graph.

* Is there an interaction between the cardinality of discrete variables and the optimality of the variance? Are there cases where the total number of blocks is _smaller_ with a larger number of variables? If so how does this affect the variance.

* It's not entirely clear to me how this algorithm will propose in practice. I would be interested to see some simulations.

* The authors talk about blocking on variables which are post-treatment, but don't discuss a critical aspect. Using a post treatment variable for blocking will change the _estimand_ itself from the total effect to a direct effect. This should be included in the dicussion.

---

### Review · Reviewer_Q7Fe · 2022-06-14

**Summary Of Contributions:**

The paper present an polynomial time algorithm to find a stable subset of covariates for blocking used in causal effect estimation. The algorithm shows 3 steps how to find the stable subset, given the underlying causal graph.

**Broader Impact Concerns:**

Not a concern

**Requested Changes:**


Detailed Comments:

1. S1.1: " .. is, confusing..." comma placement
2. "Randomized experiments work on three basic principles" is there a citation on this part (or justification if it's new)?
3. S2.1: A lot of results here make sense as blocking minimizes the variance, which in turn maximizes the precision. The results are straightforward. However, it does not say about the size of blocking. Is bigger size of blocking always better? How much is gained? If size is a concern in sample efficiency, what would be the trade-off and what is the optimal blocking size?
4. authors should make clear what is the difference between CATE and blocking.
5. "For covariates in the set Z, we need to form Z∈Z v(Z) different blocks." if you need that many blocks, it just becomes the covariate set?
6. S3.2: Notation wise, P(W) would be parents of W in any graph and cannot be distinguished from each other.
7. Problem 1: arg min Z minimizes the value of Z, not its size.
8. Example 3: V2 → V5 ← Y, i don't see this path in any figure.
9. Bi directed path definition: can be moved earlier since it is used in one of the examples in Figures.
10. "V2 is a collidor on the path V3 → V1 ← V6". I don't fully understand this part, can you explain further?
11. what algorithm do you use to find path in algorithm 1 line 9?


**Strengths And Weaknesses:**

Strength:

- Figures and examples are used to illustrate the idea
- Some theoretical results are shown.

Weakness:
- Presentation contains errors and/or are not precise in places.
- No experiment study to show how the stable set perform better than any other set or baseline adjudgments.

---

### Comment · Action_Editors · 2022-06-16
**Rebuttal**

Dear Reviewers, Thank you for the timely and insightful reviews!

We will now start the discussion period.

The Authors are encouraged to respond to the reviews.

*The goal of the discussion period is for the Reviewers to gather all the information needed to be comfortable submitting a decision recommendation for this submission within 2 weeks*.

---

### Decision · Action_Editors · 2022-07-26

**Recommendation:** Reject

**Comment:**

The manuscript provides a polynomial time algorithm for finding a set of variables to be used to form blocks when designing an experiment. The algorithm assumes that a causal graph and a treatment variable are given.

While the manuscript addresses an important problem, reviewers raised several major concerns:
1.	Since "forming blocks" and "deciding which variables are good or bad controls" is the same task it follows that the software packages cited by [1] already provide algorithms to form blocks from a given causal model.
2.	Missing related work on experimental design such as [2,3,4,5].
3.	Missing simulation studies. The manuscript should provide a set of simulations to demonstrate the relative efficacy of the proposed algorithm.

There is no author rebuttal.

References:
[1] Carlos Cinelli, Andrew Forney, and Judea Pearl. “A crash course in good and bad controls”. Available at SSRN 3689437, 2020.
[2] Rotnitzky, Andrea, and Ezequiel Smucler. "Efficient Adjustment Sets for Population Average Causal Treatment Effect Estimation in Graphical Models." J. Mach. Learn. Res. 21 (2020): 188-1.
[3] Kallus, Nathan. "Optimal a priori balance in the design of controlled experiments." Journal of the Royal Statistical Society: Series B (Statistical Methodology) 80.1 (2018): 85-112.
[4] Li, Xinran, and Peng Ding. "Rerandomization and regression adjustment." Journal of the Royal Statistical Society: Series B (Statistical Methodology) 82.1 (2020): 241-268.
[5] Harshaw, Christopher, et al. "Balancing covariates in randomized experiments with the Gram--Schmidt Walk design." arXiv preprint arXiv:1911.03071 (2019).